# Using real-world evidence to evaluate the long-term health and economic impact of the digital tool Grohealth W8Buddy supporting access to specialist weight management services: a protocol for a cohort observational study

Pranay Singh Deo [iD],[1] Amy Grove,[2] Mengxi Zhang [iD],[1] Keith R Abrams,[3,4] Peter Auguste,[5] Thomas M Barber [iD],[1,6] Tracy Gazeley,[6] Richard Green,[6] Frances Griffiths [iD],[1] Jonathan Hazlehurst,[2,7,8] Siew Wan Hee [iD],[1,9] Amit Kaura [iD],[10] Akhila Mallipedhi,[11] Sarah O'Toole,[6] Arjun Panesar [iD],[12] Nicholas Parsons,[1] Emma Scott,[1] Charlotte Summers,[12] Manreet Thind,[6] Marie Thorpe,[1] Anjali Zalin,[13] Petra Hanson [iD] [1,6,14]

For numbered affiliations see end of article.

**Correspondence to**
Dr Petra Hanson;
petra.hanson.1@warwick.ac.uk

## ABSTRACT

**Introduction** Obesity affects over a quarter of the UK population and can lead to serious health issues. NHS Specialist Weight Management Services (WMS) offer treatments including lifestyle advice, psychological support and medications, but access and availability vary by region. Although around 4 million people could be eligible for NHS Specialist WMS annually, capacity is limited to 35 000, severely limiting overall access for those who need it. While digital technology has started to be used in WMS, more evidence is needed to confirm its long-term effectiveness, acceptability and cost-effectiveness. This study explores the use of Gro Health W8Buddy, a digital platform and app providing remote Specialist WMS. It aims to determine the long-term health benefits of remote WMS pathway Gro Health W8Buddy compared with standard NHS WMS delivered in hospitals, and to improve patients access to services.

**Methods and analysis** The study is a real-world evaluation with observational data collection. We will recruit 450 study participants from four NHS specialist WMS who will choose either standard NHS WMS or the digital pathway Gro Health W8Buddy. Participants are being given the option to choose their pathway to generate real-world evidence. We will measure and analyse health outcomes including weight loss, time taken to be treated and cost-effectiveness, at 18 months and follow up at 24 months for later analysis (outside of this core funding). We will gather experiential data from patients and healthcare professionals through surveys, observation and interviews.

**Ethics and dissemination** Ethical approval has been obtained from NHS Health Research Authority (HRA) and Health and Care Research Wales (HCRW) (Supplementary Figure 3) (REC reference: 25/EM/0147). Our findings will be disseminated through academic publications, conference presentations and stakeholder engagement.

---

### STRENGTHS AND LIMITATIONS OF THIS STUDY

⇒ The study is structured around four distinct work packages, each employing a unique research methodology.

⇒ The digital intervention has been co-created with NHS staff and patients and is backed by research.

⇒ This research methodology has been shaped by patient and public involvement input from all four NHS sites.

⇒ The study methods enable real-world evidence to be generated by allowing participants to choose their own pathway of care. Therefore, we will provide credible evidence for further roll-out of digital technology in weight management services across the UK.

⇒ Allocation is not randomised, which may lead to more participants in digital care or vice versa. This approach inevitably introduces complexity into the analysis, but it is a design choice advised by patient contributors.

---

**Trial registration** ISRCTN89168871; Pre-results.

## INTRODUCTION
### Background

In the UK, a quarter of adults live with obesity, with higher rates among people from Black and Asian ethnic groups and people living in the most deprived areas.[1] People living with obesity are at increased risk of chronic diseases including type 2 diabetes mellitus (T2DM) and cardiovascular disease as well as at risk of shorter life expectancy.[2] Treatment

of obesity cost the NHS £6.5bn in 2022, with an associated ill health annual full cost of £58bn.[3] The NHS and Office for Life Sciences (OLS) identified treatment and prevention of obesity as one of its healthcare missions.[4]

A tiered approach to weight management exists in the UK. Tiers 1 and 2 are delivered in community and primary care. Tier 3 is delivered by specialist Weight Management Services (WMSs) which are often based in a hospital setting. WMS comprises multidisciplinary teams who provide dietetics, psychology and consultation by specialist doctors. Tier 4 services include bariatric surgery.[5] In the UK, National Institute for Health and Care Excellence (NICE) recommends that people with a body mass index (BMI) over 40 kg/m$^2$ or over 35 kg/m$^2$ with medical problems (eg, T2DM, hypertension) are eligible for referral to WMS.[6] However, there is substantial geographical and service variation, with many areas lacking access to services.[7]

Outcomes from traditional face-to-face WMS are mixed, with 50% of patients in most NHS WMS achieving a 5% wt loss over 6–12 months according to a recent review.[8] Retention rates are low (<55%) and there is little long-term (>12 months) data on sustainability of weight loss.[8] WMSs have limited capacity for just 35 000 patients,[9] yet 4 million people are eligible for services. This represents a significant capacity issue.[10] Due to these access constraints, only 3.13% of this population receive a referral to WMS.[9] There is also marked regional variation in service design and accessibility.[7] High-risk groups such as individuals from deprived areas in the UK often face barriers to accessing in-person Tier 3 services due to costs of transport and having support with childcare.[11] This is important to note as the obesity prevalence is 37.4% in deprived areas compared with 19.8% in less deprived areas.[12]

In March 2023, NICE recommended the obesity medication semaglutide (Wegovy (TM)) for people accessing WMS.[13] Additionally, tirzepatide was recommended for obesity treatment from December 2024.[14] These treatments represent a step change in tier 3 WMSs and are part of the government-backed pilot to widen access to this medication with the use of digital technology.[15] However, guidance on the incorporation of digital technology to address how to safely incorporate the drugs remains unclear. We aim to fill the gap with this study.

Digital technology for managing chronic disease, including obesity, is expanding rapidly in the NHS.[16] Acceptability of digital technology for weight loss is high in both community and WMS settings, which is important to note considering the current barriers to WMS uptake and accessibility.[17–20] However, effectiveness varies. Evidence shows weight loss ranging from 13% at 4 months (digital+meal replacements)[21] to 7.6% at 12 months (digital+coaching).[22] A community-based digital weight management intervention recorded 5% wt loss at 2 years in 18% of users.[20] According to the evidence, there is a gap and need for further digital weight management research to investigate standard and digital care,

alongside the integration of pharmacotherapy (eg, semaglutide) and the use of a multidisciplinary team.[21] Currently, the evidence on the digital divide created by digital technology for weight management is lacking.[23] Therefore, innovative approaches are needed in the NHS WMS to reduce health inequality, address efficiency, increase capacity and improve access to pharmacotherapy for obesity.

As a result of the above evidence, we co-created a digital support tool *Gro Health W8Buddy* (W8Buddy) with a digital company Diabetes Digital Media (DDM) to provide digital technology to deliver NHS WMS.[7] W8Buddy provides personalised WMS to support patients to achieve their self-selected health goals and support weight loss with real-time remote monitoring. The creation of W8Buddy was supported by an *HEE Topol Digital Fellowship* and *NHS Clinical Entrepreneur scheme*.[24] Evidence has shown that W8Buddy helps to reduce weight, improve glycaemic control and well-being among people living with obesity, overweight or with T2DM.[25 26] W8Buddy was endorsed by *NICE Early Value Assessment* for Digital Technologies in Specialist Weight Management in October 2023.[27]

Our initial 1-year evaluation of 412 patients showed that 220 patients using W8Buddy (version 1—information resource) alongside traditional WMS lost significantly more weight (0.75 kg/month) than 192 patients using traditional WMS.[28] Since then, W8Buddy has been adapted to provide a fully remote WMS. This study aims to provide longer term real-world evidence (RWE) of the clinical and cost-effectiveness of the adapted W8Buddy in the NHS.

### Research question

What is the long-term (18-month) impact on clinical outcomes, including health-related quality of life (QoL) and cost-effectiveness of W8Buddy with and without weight management medication and lifestyle support to people living with obesity, compared with WMS standard care?

### Objectives

1. To generate longitudinal evidence of the clinical and cost-effectiveness of the W8Buddy from four NHS WMS, University Hospital Coventry & Warwickshire (UHCW), University Hospitals Birmingham (UHB), London Luton Trust (LLT) and Hywel Dda University Health Board Trust (HDUHBT), using the digital pathway W8Buddy.
2. To generate sufficient evidence to satisfy NICE guidance requirements.[27]
3. To understand the process of implementation and use of W8Buddy within WMS.
4. To create a manual for safe incorporation of a digital pathway delivering weight management medication into existing WMS across the UK.
5. To make recommendations for how we can optimise existing infrastructure of WMS to improve ongoing RWE generation.

## METHODS

### Study design

Multisite prospective comparative observational cohort study with embedded cost-effectiveness analysis and process evaluation. Comprising four work packages to assess the following: (1) Set up, Patient and Public Involvement and Engagement (PPIE) and Stakeholder engagement, (2) Clinical impact, (3) Health economic impact and (4) Process evaluation.

### Eligibility criteria to enter WMS in the UK

Inclusion criteria reflect the eligibility of people to enter the clinical services participating in this research, including services with additional eligibility criteria restrictions (subject to local processes that may differ slightly). However, while individual services may have further restrictive criteria, which is reflective of many WMSs, all participants are included within:

► Adults aged 18 years and above
► Class 2 obesity (BMI ≥35 kg/m$^2$) with at least 1 medical complication (eg, pre-diabetes, T2DM); or patients with obesity class 3 (BMI ≥40 kg/m$^2$), without any complications.

The eligibility criteria for access to Tier 3 WMS for adults from South Asian, Chinese, other Asian, Middle Eastern, Black African or African-Caribbean background have BMI cut-offs of 37.5 kg/m$^2$ without comorbidities.

### Exclusion criteria

► Pregnant or breastfeeding.
► T1DM.
► End-stage renal disease (eGFR <15, receiving renal replacement therapy).
► Active or untreated alcohol or drug dependency.
► Decompensated liver disease.
► History of anorexia, undergoing treatment or awaiting treatment for DSM-V classified eating disorder.
► Severe mental health disease, including acute mental health crisis or self-harm in the last 12 months.
► Unstable mental health disease, including unstable personality disorder, schizophrenia and bipolar disorder.
► Unwilling or unable to commit to participation across the length of the research study.
► Patients targeting bariatric surgery within the length of the research study.
► Participating in another weight management research intervention.
► Participating in the diabetes remission programme.

### Study setting and sites

This study will run at four NHS sites (UHCW, UHB, LLT, HDUHBT). Standard care differs across the recruiting sites, including the mixed provision of obesity medication. Sites without medication as standard care will be noted as the digital pathway without medication.

### Patient and public involvement

A W8Buddy PPI Group has been established with a minimum of four PPI Representatives with lived experience of WMS from each site. Potential PPI representatives have been identified through WMS teams at each participating site and obesity charities. Those interested in joining the groups are invited to meet with the PPIE Lead and/or UHCW PPIE Team to discuss the opportunity, commitment required and support offered and to find out about their experience and background. PPI representatives selected represent the diverse population of the WMS users in each area. The Group meets quarterly to ensure ongoing input from people with lived experience, including identifying outcome measures that are meaningful for people with obesity, providing feedback on participant-facing documents, working on creating local pathways and training materials. The PPI Group will also shape analysis and risk mitigation, may take part in conducting semistructured interviews as part of qualitative research, and support dissemination of findings afterwards. This research methodology has been shaped by PPI input from all four NHS sites.

### Sample size calculation

We will collect weight data at baseline and a minimum of three occasions during the study, for both pathways. Powering the study to show that the digital pathway is non-inferior to the standard of care pathway, with a difference less than 4 kg, at 90% power and a two-sided 5% level, gives a sample size of approximately 180 participants. Further assuming a correlation between the repeated weight measurements in the range 0.2–0.5, which previously collected data suggests is reasonable,[28] based on a longitudinal mixed model/repeated measures Analysis of covariance (ANCOVA) gives a total sample size of 360. Conservatively assuming 20% loss during follow-up, the target sample size is 450.

Given the current widespread use of the digital (W8Buddy) intervention at the lead centre (UHCW), the control (standard of care; SC) intervention at this site will be different from that offered at the other sites. For this reason, we plan to characterise the control intervention at UHCW as a 'hybrid' digital intervention. We will aim to recruit sufficient participants (ie, 360) at the three other sites (UHB, LLT, HDUHB) to fully power the study for the planned comparison between the digital and standard of care groups. Additionally, we will collect data from the 'hybrid' digital intervention at UHCW to augment and further refine the primary analysis.

### Screening, recruitment and consent

Potential participants will be identified by research nurses once they have been referred (or self-referred) to the local Tier 3 WMS. Participants will be screened electronically against pre-specified criteria (figure 1).

Eligible participants will be able to choose how they receive their WMS—either in the standard NHS delivery, or via fully digital platform Gro Health W8Buddy.

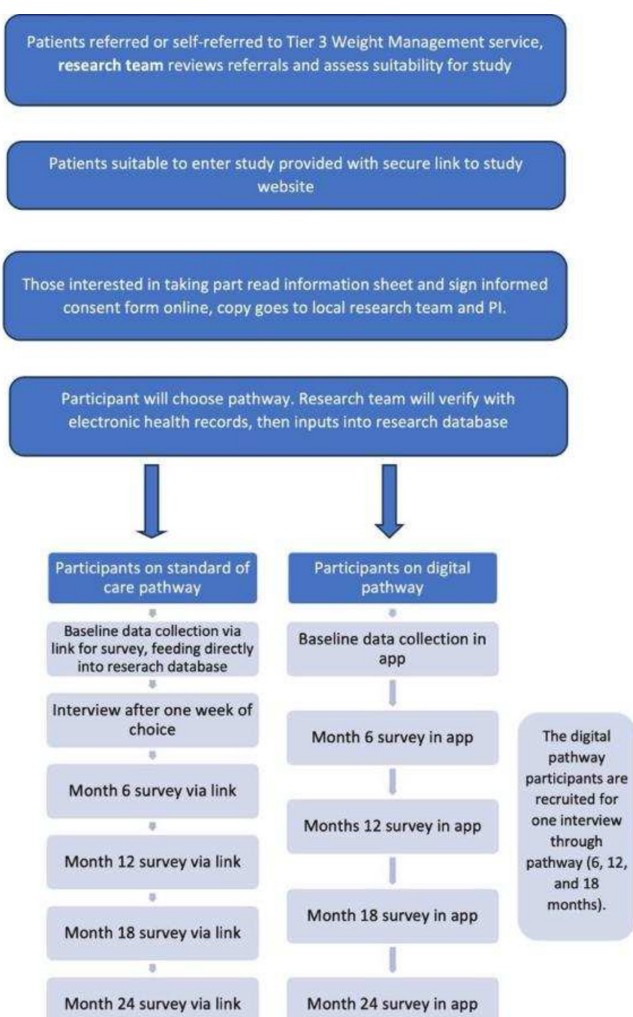

**Figure 1** Participant flow chart. The flow chart describes where participants will be providing data in both the digital and standard of care arms. PI, principal investigator.

Site-specific standard of care pathways will be delivered by the usual healthcare professionals, employed by each NHS site. The digital pathway will be delivered by healthcare professionals employed by DDM (providers of Gro Health W8Buddy). We aim to recruit a total of 450 participants across both arms from September 2025 to February 2026.

Those who satisfy the study inclusion criteria will be contacted by the research team via telephone. Brief information about the study will be provided to potential participants, including a link to a website with information about the study. This website will be password protected as only eligible participants will be able to view it. Digital information sheets and consent forms will be hosted on the website.

If a patient is unable to access the website (therefore, unable to complete electronic consent), the local site research nurse will invite them to the hospital. They will show the participant the research website and help them with electronic consent. Later, the research nurse will print out the consent form (signed copy) only for the patient if they do not have access to email.

Electronic consent for data collection, with an option to consent for ongoing data collection, will be obtained. Following informed consent, participants will be able to choose the digital or standard care pathway via the website. A copy of the signed consent form will be sent to the participants, local research team and site principal investigator (PI).

To support data collection in the standard care pathway, participants will be offered a £10 voucher for each completed survey at months 0, 6, 12, 18 and 24; vouchers will not be provided for sign up only, surveys must be completed by participants.

### Intervention Gro Health W8Buddy
Gro Health W8Buddy is a virtual specialist WMS pathway. It is accessed through a digital behavioural change platform called the Gro Health app (see, figure 2). Participants on the W8Buddy digital arm will receive 52 weeks of support from healthcare professionals employed by DDM. At week 52, each participant on the digital pathway will be assessed and the following options offered: discharge to primary care, onward referral to local tier 4 if interested in surgery, onward referral for local tier 3 MDT discussion, extension of digital pathway by 6 months, referral to local tier 2 WMS.

### Standard care
Participants in the standard care arm will receive the tier 3 WMS that is delivered by their local NHS trust. All four NHS trusts included in the research have differences in their tier 3 service, which will be explored in detail as part of the study. The differences range from the amount of input from healthcare professionals (psychologist, dietitian, physician), modality of sessions (1–1 or group sessions) and frequency. Post study care for participants in standard care will be delivered in accordance with their NHS trust policies for tier 3 wt management. As discussed previously, the W8Buddy intervention is included in the standard of care at the lead centre (UHCW). It is used as a digital support tool only (no access to clinicians via W8Buddy), enhancing the usual face-to-face care, and as such, differs from entirely remote digital pathway. For this reason, the model of delivery of care is hybrid at UHCW (refer to table 1 for further detail).

### Assessment and management of risk
Risks relate to data management/reporting across four sites and management of the digital pathway delivered by DDM. The latter is assured by DDM. Table 2 lists these. Further information on data management and flow can be found in online supplemental table 2, adverse events (online supplemental table 2) and online supplemental figure 1.

### Outcomes
The primary outcome is changes in body weight after 18 months of using either (1) a fully digital weight management programme (remote digital pathway Gro Health W8Buddy) or (2) site-specific delivered weight

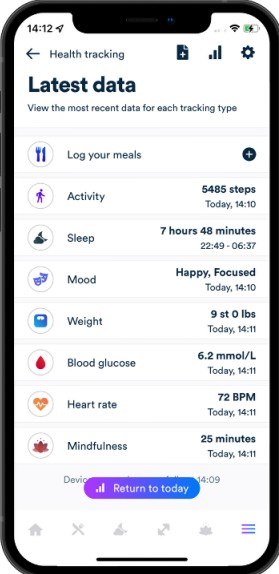
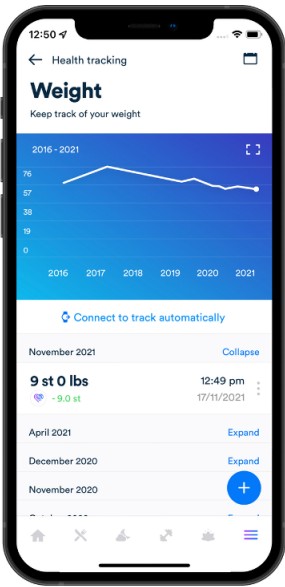
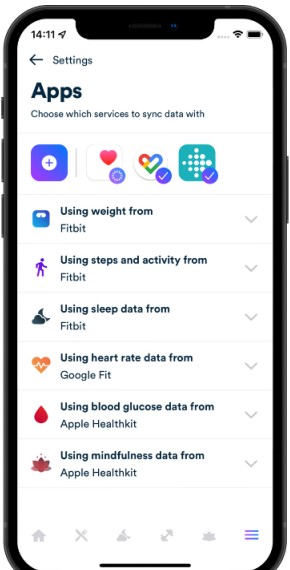
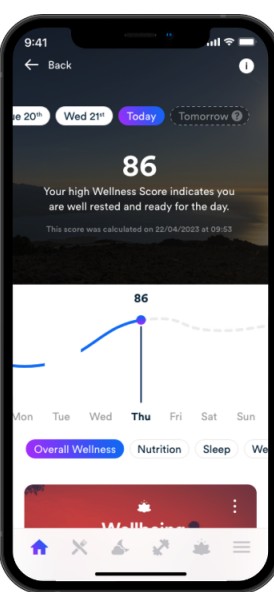

**Health dashboard**  **Weight data**  **App syncing**  **Wellness Score**

**Figure 2** Gro Health W8Buddy platform and app—description and visual representation of the W8Buddy platform in an app format.

management programmes incorporating face-to-face and/or virtual group sessions (standard care).

Secondary outcomes are metabolic markers (HBA1c, lipids), blood pressure, waist circumference and health-related QoL (EQ-5D-5L).[28]

### Data collection
In online supplemental table 1, we detail what baseline and follow-up data will be collected to assess primary and secondary outcomes. During the economic evaluation, additional cost and process evaluation qualitative data will be collected.

### Monitoring
The study will be monitored by the study management group and the University of Warwick representative of the Sponsor to ensure that the study is being conducted as per protocol, adhering to Research Governance and Good Clinical Practice (GCP). Central monitoring activities will be performed such as data quality checks. The recruiting site is obliged to assist the sponsor in monitoring the study. These may include hosting site visits, providing information for remote monitoring, or putting procedures in place to monitor the study internally. The approach to, and extent of, monitoring will be specified in a study monitoring plan determined by the risk assessment undertaken prior to the start of the study.

**Table 1** Standard of care

| NHS site | Standard of care | Weight management medication offered in current tier 3 service* |
|---|---|---|
| UHCW | Hybrid standard of care with digital W8Buddy input and in person MDT support. Mixture of one-to-one and group sessions delivered by dietitians, psychologists and consultant endocrinologists. | No |
| UHB | Mixture of group sessions and one-to-one appointments delivered by dietitians, psychologists and consultant endocrinologists following a face-to-face introductory group session. | No |
| LLT | Mixture of face-to-face and group sessions delivered by psychologists, dietitians and consultant endocrinologists. | No |
| HDUHBT | Mixture of face-to-face and group sessions delivered by psychologists and dietitians. After completion, they are seen by a consultant. | Yes |

*Please note this information is accurate at time of publication.
HDUHBT, Hywel Dda University Health Board Trust; LLT, London Luton Trust; MDT, multidisciplinary team; UHB, University Hospitals Birmingham; UHCW, University Hospital Coventry & Warwickshire.

**Table 2** Assessment and management of risk

| Data reporting | Not reporting data by participants, mitigated by regular support via the app or contact from the research team if not completing surveys. |
|---|---|
| Data protection | As with any online service, proper data handling is assured by high security data storage in line with NHS Digital policies. Data from NHS sites transferred to analytical partner using secure pathways. |
| Delivery | The site research team will oversee all participants, including those accessing digital pathway. The first contact for any queries for participants on the digital pathway will be the digital provider, and if the query is not resolved, the research site team will be contacted. The project will be managed by PH and AG via weekly WP meetings between sites and project manager, with monthly oversight by each site PI. Three-monthly meetings of the PPIE group will ensure ongoing input from patients. Annual project steering committee meetings will review risk registers. |
| Imbalanced recruitment | More patients opting for digital rather than standard of care pathway (or vice-versa), mitigated by over-recruiting standard of care (or digital) patients and adjusting as per analysis plan using propensity-weighted approach, purposive sampling and ongoing monitoring of recruitment, communication to sites, telephone and offline packs as another option for those choosing digital pathway. |
| Engagement with the digital pathway | Low engagement from high-risk/target groups resulting in lower enrolments and participation from those most in need of the service. Mitigated by patient-friendly resources, onboarding support and a research nurse at each site to act as a point of call. |
| Patient safety | Without face-to-face contact, healthcare professionals may miss important physical signs and symptoms that might indicate a health concern for study participants. Participants might also misunderstand or misuse instructions regarding their medication, increasing the risk of side effects or adverse events. This is mitigated through symptom tracking, face-to-face videoconferencing/virtual calls, identifying signs of body language and recording sessions for audit to ensure ongoing patient safety. Digital phenotyping of patients occurs in real-time, and any thresholds of anxiety or depression are noted in the clinical dashboard and the clinical team are notified. Omni-directional communication through the platform allows clinicians to engage directly with patients and vice versa with up-to-date, real-time data. Patient expectations are managed by informing the patient if their coach is offline and encouraging them to seek emergency medical assistance if at risk. |
| Medication safety on the digital pathway | Prescribing medication remotely requires a comprehensive understanding of the patient's current health status, medications and lifestyle, which can be challenging to obtain in a virtual environment. There is also the risk of potential drug interactions that might be overlooked without a detailed in-person evaluation. This is mitigated through patient completion of psychological assessments, medication eligibility assessment and approval for medication from the multidisciplinary team (MDT) providing the patient's care. Regular reviews ensure medication is not mismanaged and appropriate for the patient. Patient is provided with videos and resources on how to inject. |
| Psychological safety on digital pathway | Weight management is closely tied to self-esteem and body image, which can be negatively impacted if the focus is solely on weight loss. Additionally, the digital nature of virtual services can exacerbate feelings of isolation and lack of support. This is mitigated through support from a psychotherapist/psychologist throughout the programme, access to private/group coaching for on-demand support, and artificial intelligence algorithms that provide digital phenotype measurements for a patient's likelihood of anxiety, stress and depression based on conversations/engagements in the community. |
| Eating disorders | Weight loss can trigger or exacerbate eating disorders. For individuals with a history of eating disorders, weight loss medications could potentially be misused in a harmful way. This is mitigated through patient completion of psychological assessments, medication eligibility assessment and approval for medication from the MDT team providing the patient's care. On the digital pathway, regular reviews with a pharmacist and psychologist ensure medication is not mismanaged and appropriate for the patient. Signposting and referral back to the GP will be provided to any patients who display signs of disordered eating. On the standard of care pathway, local protocols will be followed. |
| Engagement with the standard of care pathway | High drop out from the standard of care pathway: there is a high drop out from the standard of care service in tier 3 Weight Management, resulting in high loss to follow-up. This will be mitigated by having a dedicated research nurse who will contact patients to ensure they complete all data collection in a timely fashion. Additionally, participants in the standard of care will receive vouchers to compensate them for the additional surveys they will be asked to do. |

GP, general practitioner; PI, principal investigator; PPIE, Patient and Public Involvement and Engagement; WP, work packages.

## Analysis

### Statistical analysis plan

A detailed statistical analysis plan will be prepared by the statisticians and agreed with the study team, prior to final analysis, and which will follow the NICE RWE Framework.[29] Comparisons will be primarily on an as-started (intention-to-treat, ITT) basis. We do not expect any or only minor levels of treatment switching or discontinuation for the W8Buddy group. Therefore, we would expect the on-treatment and as-started treatment effect estimates to be equivalent in this population. We will report the result according to the Strengthening the Reporting of Observational Studies in Epidemiology guideline.[30]

### Summary of baseline data

Demographic and baseline data will be summarised overall and by treatment arms and sites. Categorical data will be summarised as count and percentage, and continuous data will be summarised as number of missing data, mean with SD, and median with 25th and 75th percentiles.

### Missing data

Patterns of missing data/discontinuation will be reported and assessed if missingness was at random or provided useful information on treatment efficacy.

### Primary and secondary outcome analysis

The primary analysis will compare temporal changes in weight for the W8Buddy and standard care group, from baseline to completion of follow-up (at 18 months), using longitudinal mixed effects regression models. The effectiveness of the W8Buddy will be quantified by differences in rates of weight change compared with standard of care during follow-up. All the secondary outcomes will be analysed individually with longitudinal mixed effects models.

For all analyses, we will consider adjusting for a range of demographics and clinical characteristics (eg, age, gender, ethnicity, Index of Multiple Deprivation/Welsh Index of Multiple Deprivation, comorbidities, NHS site, mode and distance of travel and other confounders identified using Directed Acyclic Graphs with stakeholders PPI members and members of the steering committee)[31] using propensity-weighted longitudinal mixed models. We expect the fixed effects to be demographics and clinical characteristics as mentioned above and the random effects to be the sites and individuals. A sensitivity analysis using distance to NHS site as an instrumental variable[32] will also be conducted to assess the assumption of no unmeasured confounders.

The comparison between the hybrid intervention and both the standard of care and fully digital intervention will be an unpowered secondary analysis, that we hope will offer additional insights and help to characterise the magnitude of intervention effects from the primary analysis.

### Economic evaluation

We will develop a costing toolkit to provide transparent resource use and costs for the delivery of the standard care and digital delivery pathways. The toolkit development process will comprise three stages: planning, data collection, synthesis and analysis.[33] [34] We will require resource use and cost estimates for delivery of W8Buddy providing weight management medication and standard of care (Tier 3 WMS). We will produce a systematic review of cost-effectiveness literature to inform the development of a decision-analytic model. The model will be used to estimate the cost-effectiveness of Tier 3 WMS in addition to W8Buddy compared with WMSs delivering weight management medication to support people with obesity. This will be fundamental to help to understand the long-term cost-effectiveness of weight management programmes as they allow for the extrapolation of the short-term benefits of interventions.

### Long-term follow-up analysis

Outcomes collected beyond the 18-month primary outcome will be analysed at a later stage, subject to further funding. The aim of the longer-term follow-up is to explore the long-term impact of digital weight management interventions. We expect the analyses will be done similarly to our primary and secondary clinical and health economic analyses.

### Process evaluation

A mixed methods process evaluation will be conducted to understand the implementation of W8Buddy. Process evaluation data are listed in online supplemental table 1.

Electronic data from intervention and control pathways: We will collate data from the electronic clinical records, intervention and study records of all study participants on:

► Fidelity: Delivery of coaching sessions, check-ins and physical activity sessions and progress reviews.
► Adherence: Including biomedical explanations of adherence (comorbidities), we will assess user adherence to each consultation in the digital pathway (a patient attending 75% of consultations or more is considered adherent).
► Who chooses the digital pathway or not.
► Who engages with the digital pathway, their level of engagement and predictors of engagement.

### Qualitative data collection and analysis

We will record observation data as field notes and collate documents. Qualitative data will be obtained from participants' interviews via telephone or videoconference. We will audio record the interviews, transcribe using software and check the transcripts against the audio recording. We will undertake thematic analysis, with independent quality checks.[35] This analysis will be undertaken independently of the quantitative analysis team.

### Initial implementation and health professional experience of delivering digital pathway

We will observe site team meetings and implementation of the intervention, access team documents and undertake interviews with team members implementing the digital

pathway to understand how they are meeting implementation challenges, training delivered and quality assurance processes put in place. During the observations, the patient will be asked for permission for the researcher to remain present during the health professional-patient interaction. If this is refused, the researcher will leave.

In months 12 and 18, we will interview those delivering the digital pathway to understand their experiences, how they adapt WMS and any unexpected pathways or consequences (estimate 5/site). We will use implementation frameworks[36] to guide thematic analysis of implementation data while remaining alert to new themes. Early analysis of data on implementation will be used to provide formative feedback for subsequent intervention site teams.

### User experience of digital pathway

In semistructured interviews with users, we will explore their experiences of engaging with the digital pathway, influences on their engagement, challenges and how these were met, adaptations, impact and mechanisms of impact (estimate 10/site, total 40).

We will undertake brief structured interviews with participants not choosing the digital pathway (5/site) and participants not adherent to the digital pathway and/ or medicine (5/site). We will ask participants why they did not choose the digital pathway/drop out or return to the digital pathway/medicine. We will develop our brief structured questions with input from the study PPI group to ensure neutral, non-judgemental phrasing. We will sample participants consecutively across study sites and approach them for recruitment at least a week after their decision.

### Describing the control intervention patient pathway

A researcher will visit each site to observe the control intervention. The team providing the control intervention will provide a written description of the control intervention patient pathway. From observation and conversation, the researcher will add additional details and variation on that pathway. The study team will analyse the observation notes to compile a description of the control intervention and check this with staff delivering the control intervention. Study sites will extract from the appointment system and relevant clinical records of their control intervention pathway, patient adherence data in terms of attendance at appointments, whether they are individual or group and duration of appointments, and use of medication.

### Ethics and dissemination

Ethical approval for this study has been obtained from the Health Research Authority (HRA) Research Ethics Committee (REC) and Health and Care Research Wales (HCRW) (REC reference: 25/EM/0147) (online supplemental figure 3). Informed consent will be required to enter the study, as well as to share anonymised data with local healthcare team and study team. Participants will be consented to long-term follow-up using routinely collected data. Protocol amendments (if any) will be reported and discussed with National Institute for Health and Care Research (NIHR) and local ethics committee.

Study data will be collected on the screening logs, W8Buddy App and validated questionnaires (see online supplemental table 1) electronically. Health record data for each site will be accessed by the local research team. An Electronic Data Capture system will be used to record and store study data transferred securely from the W8Buddy App and hospital medical records. This will be the main Study Database. Only research team members and members of the DDM team will have access to it. They will be listed in the delegation log and must hold GCP certification. Audio recordings of interviews will be stored on secure University of Warwick computers and accessed only by the research team located at Warwick.

We have a tiered approach to dissemination to ensure our findings reach participants, policy makers and practitioners. We will host a stakeholder engagement dissemination event at the end of the study to maximise the reach of our findings.

Our results will inform UK guidelines (incorporation of digital tool delivering pharmacotherapy into obesity management guideline). Academic publications will include a protocol paper, main results of RWE evaluation, the health economic assessment and a paper describing the scale-up in NHS services and impact on multiple long-term conditions. We will create an NHS business case, build on ongoing partnership developed across four adoption sites, produce equality and health inequality impact assessment, and have evidence for health and financial impact to support NHS adoption. We will improve NHS services by facilitating the testing and scale-up of our digital innovation which can support other Trusts wanting to deliver digitally enabled services. This will include recommendations to other service providers, targeted education and creation of a virtual learning environment (hosted by an established Warwick University platform). Our study outputs will provide evidence to apply for a NICE review of clinical guidelines and an update to NHS reimbursement criteria, which will open the door to national adoption by WMSs.

## DISCUSSION

Research is needed to determine how we can most effectively deliver WMS to more people.[21] One potential solution to address issues of patient access is the introduction of digital services.[14–18] However, long-term evidence about clinical and cost-effectiveness is required before digital pathways are fully integrated into NHS WMS across the whole country.

In this study, we aim to provide longer-term RWE of the clinical and cost-effectiveness, as well as barriers and facilitators of uptake of W8Buddy digital pathway implemented in four NHS sites. All these potential findings are essential considering the current data on the demand for WMS, as well as the health and economic impact of obesity.[2 3] Guidance

on the incorporation of weight management drugs alongside digital interventions is also an unclear gap we intend to fill with our research. With capturing the variation that exists among our four NHS sites, we will provide guidance on implementation into further NHS sites.

Allowing study participants to choose their own pathway will allow RWE to understand uptake of the digital pathway when incorporated into standard care. Furthermore, we are assessing whether W8Buddy can decrease NHS waiting times and costs. Individuals living with obesity could, in time, be supported virtually without the need to attend in person specialist services with the associated costs and challenges of access. Investigating pathways, waiting times and accessibility is important for obesity research considering the low retention rates and high demand but low availability in WMS.[7 11 12] The use of a digital pathway will also provide further insight for innovative diagnostic pathways for obesity management, such as moving from acute to home care and being offered to more patients in a cost-effective way, which would further contribute to the literature base.

We acknowledge the challenges that may occur with the adoption of a digital pathway into NHS sites such as recruitment rates, levels of engagement and differences in WMS at NHS sites. However, capturing any challenges that may arise and providing solutions will benefit the future implementation of W8Buddy and contribute to generalisable knowledge. Our design means that we will not have matched cohorts across treatment arms, meaning both known confounders (eg, age, BMI) and unknown factors may influence outcomes. To address this, our planned analyses will incorporate statistical adjustments to account for these differences as far as possible. We recognise the limitations of this approach, but it is consistent with the pragmatic design and will allow us to report findings that are representative of real-world practice.

We aim to improve NHS Specialist WMS services by facilitating the evidence generation and scale-up of digital innovation Gro Health W8Buddy. This could, in turn, support other NHS organisations wanting to deliver digitally enabled services. The findings of this study will be used to generate recommendations for other service providers, targeted education and creation of a virtual learning environment. It is aimed that the study outputs will provide evidence needed for full NICE technology appraisal, as well as evidence for NHS commissioners to facilitate national adoption of technology in specialist WMS.

**Author affiliations**
[1]Warwick Medical School, University of Warwick, Coventry, UK
[2]University of Birmingham, Birmingham, UK
[3]Department of Statistics, University of Warwick, Coventry, UK
[4]Centre for Health Economics, University of York, York, UK
[5]School of Social Policy and Society, University of Birmingham, Birmingham, UK
[6]University Hospitals Coventry and Warwickshire NHS Trust, Coventry, UK
[7]Birmingham Health Partners, Birmingham, UK
[8]Department of Diabetes and Endocrinology, University Hospitals Birmingham NHS Foundation Trust, Birmingham, UK
[9]Research and Development, University Hospitals Coventry and Warwickshire NHS Trust, Coventry, UK
[10]Faculty of medicine, Imperial College London, London, UK
[11]Hywel Dda Health Board, Carmarthen, UK
[12]DDM Health, Coventry, UK
[13]Bedfordshire Hospitals NHS Foundation Trust, Luton, UK
[14]South Warwickshire NHS Foundation Trust, Warwick, UK

**Contributors** PH (chief investigator and guarantor) initiated the research project, wrote the grant application and developed the protocol. AG (chief investigator) initiated the research project, wrote the grant application and developed the protocol. ES and MThi (project managers) supported writing the grant application, securing funding and developing the protocol. PSD (author) and MZ (post-doctoral researchers) supported writing the protocol and are involved in conducting and analysing the research. MTho, KA, PA, FG, AK, NP and SWH (all coapplicants) were involved in the methodology and analysis of the research, and also supported developing the protocol and grant applications. TH, RG and SO'T (coapplicants) supported PPI involvement, grant applications and writing the protocol. TB, JH, AM and AZ (coapplicants, all site principal investigators) supported the grant application and protocol. AP and CS (coapplicants) supported developing the protocol and producing the W8Buddy digital app.

**Funding** This study/project is funded by the NIHR (grant number NIHR208100 https://fundingawards.nihr.ac.uk/award/NIHR208100). The University of Warwick will act as study sponsor. The study will be conducted in accordance with Sponsor's Standard Operating procedures. The University of Warwick has a specialist insurance policy in place which would operate in the event of any participant suffering harm as a result of their involvement in the study Zurich Municipal Insurance. NHS indemnity operates in respect of the clinical treatment that is provided. Further information can be found in online supplemental table 2. This study is funded by the National Institute for Health and Care Research Invention for Innovation (NIHR i4i) programme (NIHR208100), (online supplemental figure 2) displays the design and management of the trial are entirely independent of the funder.

**Disclaimer** The views expressed are those of the author(s) and not necessarily those of the NIHR or the Department of Health and Social Care.

**Competing interests** PH led the co-creation of W8Buddy on behalf of UHCW. She is not a stakeholder in DDM and does not have any employment contract with DDM. She is a named inventor at UHCW for W8Buddy. She has received honoraria for delivering talks from Sanofi, AstraZeneca and Royal College of Emergency Medicine. TMB has received funding for investigator-initiated research from Bayer, AstraZeneca, Shire and TDeltaS, and honoraria for educational talks and attendance at advisory board meetings for Sanofi, Novo Nordisk, AstraZeneca, Boehringer Ingelheim, MSD, Lilly and Napp. KH is a member of the National Institute for Health and Care Excellence (NICE) Diagnostics Advisory Committee, the NICE Decision and Technical Support Units and is a National Institute for Health Research (NIHR) Senior Investigator Emeritus (NF-SI-0512-10159). He has served as a paid consultant, providing unrelated methodological and strategic advice, to the pharmaceutical and life sciences industry generally, as well as to DHSC/NICE, and has received unrelated research funding from Association of the British Pharmaceutical Industry (ABPI), European Federation of Pharmaceutical Industries & Associations (EFPIA), Pfizer, Sanofi and Swiss Precision Diagnostics/Clearblue. He has also received course fees from ABPI and the University of Bristol and is a Partner and Director of Visible Analytics Limited, a health technology assessment consultancy company. AP is the CEO and Head of AI and Ethics at DDM Health, Honorary Associate Professor at Warwick Medical School, University of Warwick, and serve as an Advisor to the Information School at the University of Sheffield, EPSRC Future Blood Testing Network, Science for Africa Foundation, NNEdPro Global Centre for Nutrition and Health in Cambridge and mentor at the Global Business School for Health at University College London. He is actively involved in the development and commercialisation of AI-based tools for digital screening, behaviour change and self management. AM is the Chief Medical Officer for DDM Health, but his involvement in the project is as an academic at Imperial College London. DDM Health will not be 36 paying AM for his involvement, nor will they receive any project funds for his involvement. JH declares advisory work previously for Novo Nordisk as well as honoraria for speaking engagements from Novo Nordisk and Eli Lilly and support for attendance at academic meetings from Novo Nordisk as well as Eli Lilly as well as academic funding from the NIHR. No other members of the research team have declared any conflicts of interest.

**Patient and public involvement** PPI contributors were involved in the design of the research. Please refer to the Methods section for further information

**Patient consent for publication** Not applicable.

**Provenance and peer review**  Not commissioned; externally peer reviewed.

**ORCID iDs**
Pranay Singh Deo https://orcid.org/0000-0001-5704-036X
Mengxi Zhang https://orcid.org/0000-0002-2815-1391
Thomas M Barber https://orcid.org/0000-0003-0689-9195
Frances Griffiths https://orcid.org/0000-0002-4173-1438
Siew Wan Hee https://orcid.org/0000-0002-0415-263X
Amit Kaura https://orcid.org/0000-0002-6962-3199
Arjun Panesar https://orcid.org/0000-0001-7050-9038
Petra Hanson https://orcid.org/0000-0002-6845-1049

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
