## [Reviewer comments · BMJ Open]

ARTICLE DETAILS

Title (Provisional)

A protocol for a cohort observational study using real-world evidence to evaluate the long-term health and economic impact of a digital tool Grohealth W8Buddy supporting access to specialist weight management services

Authors

Deo, Pranay Singh; Grove, Amy; Zhang, Mengxi; Abrams, Keith R; Auguste, Peter; Barber, Thomas; Gazeley, Tracy; Green, Richard; Griffiths, Frances; Hazlehurst, Jonathan; Hee, Siew Wan; Kaura, Amit; Mallipedhi, Akhila; O'Toole, Sarah; Panesar, Arjun; Parsons, Nicholas; Scott, Emma; Thind, Manreet; Summers, Charlotte; Thorpe, Marie; Zalin, Anjali; Hanson, Petra

VERSION 1 - REVIEW

Reviewer	1
Name	Dale, Megan
Affiliation	Cardiff and Vale University Health Board
Date	10-Nov-2025
COI	None

The study addresses issues that are clearly important, as explained in the introduction and the protocol sets out how evidence gaps identified by the NICE Early Value Assessment would be addressed. There are a number of clarifications that could be made.

It is a complex design and it would be helpful to have a table or diagram that summarises which sites are offering which intervention and comparator, as well as a clearer description of what is intended by a hybrid intervention (as used in UHCW).

The intervention is offered for 12 months at which point switching is possible for 6 months (possibly?) and then data is collected again at 24 months. It could be clearer what treatment is available for both arms for the 12-18 month and 18-24 month periods. The primary outcome is for 18 months but there is no mention of analysis of 24 month data.

Offering participants a choice of intervention is an unusual approach and different to study designs suggested in the NICE evidence generation plan. It would be helpful to explain further why this approach was chosen and how it will impact the analysis and interpretation

of results, given there may be many factors that drive that patient choice and these may impact on outcomes.

Will results be presented separately for sites with and without weight management medication?

The authors discuss the risk of low engagement from high-risk/target groups. While appreciating that the word limit may mean this information could not be included in detail, there are several mentions of issues in WMS of enrollment and retention and therefore this may warrant more discussion and engagement.

VERSION 1 - AUTHOR RESPONSE

1. Add a table or diagram to clarify the complex design:  ○ Which sites offer which intervention and comparator? ○ What is meant by a “hybrid intervention” at UHCW? - Comment 5: Include a table or diagram summarising site-specific interventions and clarify the hybrid model at UHCW. 	Page 14	Thank you, we have included this suggestion. As this is a real world study it is possible that the site provision changes as the study progresses. The final service delivery model at the end of the study will be reported in the final study manuscript. We will include detailed descriptions of any service changes that occurred during the duration of the study.
2. Clarify treatment availability across timepoints:  ○ What is available at 12–18 and 18–24 months? ○ Is there analysis of 24-month data? Comment 6: Clarify treatment options across all timepoints and whether 24-month data will be analysed.	Page 12, 13, 14 & 28	The intervention is offered for 12 months at which point switching is possible for 6 months (possibly?) and then data is collected again at 24 months. We only allow switching to standard care and it doesn't reset the participants journey It could be clearer what treatment is available for both arms for the 12-18 month and 18-24 month periods. This is not a trial and we are not allocating treatment. It has been specified after the tier 3 service is delivered in both digital and standard care arm what the participant will receive.

		The primary outcome is for 18 months but there is no mention of analysis of 24 month data. Thank for your feedback, we have now specified that 24 month data is for optional follow up (outside of this core funding) Also included for further clarity- Outcomes collected beyond the 18-month primary outcome will be analysed at a later stage, subject to further funding.
3. Justify the choice-based design:  ○ Why was participant choice included? ○ How will this affect analysis and interpretation? Comment 7: Explain rationale for allowing participant choice and its implications for analysis.	Page 4 & 28	The choice has been shaped by PPI – we have included more detail on this to make it clearer to the reader. The implications for the analysis are difficult to quantify, as currently we don't know what choices people will make and who those people are. Our planned analyses, namely, propensity-weighted longitudinal mixed effects models and directed acyclic graphs, will attempt to adjust for all possible known confounders between treatment groups.
4. Clarify whether results will be stratified by sites with and without weight management medication. Comment 8: State whether results will be presented separately for sites with and without weight management medication.	Page 28	We will use longitudinal mixed effects model for our primary and secondary outcomes. Sites is one of the random effects in the model. Hence, there will be no separate stratification analysis by sites. Treatment arm is the variable of interest in our analysis. Therefore, results will be reported by treatment arms. We will explore comparison between with and without weight management medication. However, our study is not powered to investigate this comparison.

9. Expand on engagement challenges:  ○ Especially for high-risk or target groups ○ Reference known issues in WMS enrolment/retention Comment 9: Expand discussion on engagement challenges, particularly for high-risk groups, referencing WMS issues.	Page 5 & 6	We have included the following sections of text - In the UK, a quarter of adults live with obesity, with higher rates among people from Black and Asian ethnic groups and people living in the most deprived areas. However, there is substantial geographical and service variation, with many areas lacking access to services. There is also marked regional variation in service design and accessibility. High risk groups such as individuals from deprived areas in the UK often face barriers to accessing in person Tier 3 services due to costs of transport, physical ability and having support with childcare (37). This is important to note as the obesity prevalence is 37.4% in deprived areas compared to 19.8% in less deprived areas. Acceptability of digital technology for weight loss is high in both community and WMS settings, which is important to note considering the current barriers to WMS uptake and accessibility.
---	-----------------------	---

VERSION 2 - REVIEW

Reviewer **1**

Name **Dale, Megan**

Affiliation **Cardiff and Vale University Health Board**

Date **09-Dec-2025**

COI

Thank you for addressing the comments, I have no further points.